# G-protein-coupled receptor signaling and polarized actin dynamics drive cell-in-cell invasion

**Vladimir Purvanov, Manuel Holst, Jameel Khan, Christian Baarlink, Robert Grosse***

Institute of Pharmacology, University of Marburg, Marburg, Germany

**Abstract** Homotypic or entotic cell-in-cell invasion is an integrin-independent process observed in carcinoma cells exposed during conditions of low adhesion such as in exudates of malignant disease. Although active cell-in-cell invasion depends on RhoA and actin, the precise mechanism as well as the underlying actin structures and assembly factors driving the process are unknown. Furthermore, whether specific cell surface receptors trigger entotic invasion in a signal-dependent fashion has not been investigated. In this study, we identify the G-protein-coupled LPA receptor 2 (LPAR2) as a signal transducer specifically required for the actively invading cell during entosis. We find that $G_{12/13}$ and PDZ-RhoGEF are required for entotic invasion, which is driven by blebbing and a uropod-like actin structure at the rear of the invading cell. Finally, we provide evidence for an involvement of the RhoA-regulated formin Dia1 for entosis downstream of LPAR2. Thus, we delineate a signaling process that regulates actin dynamics during cell-in-cell invasion.

*For correspondence: Robert.grosse@staff.uni-marburg.de

**Competing interests:** The authors declare that no competing interests exist.

**Reviewing editor**: W James Nelson, Stanford University, United States

## Introduction

Entosis has been described as a specialized form of homotypic cell-in-cell invasion in which one cell actively crawls into another (*Overholtzer et al., 2007*). Frequently, this occurs between tumor cells such as breast, cervical, or colon carcinoma cells and can be triggered by matrix detachment (*Overholtzer et al., 2007*), suggesting that loss of integrin-mediated adhesion may promote cell-in-cell invasion. This is further supported by the fact that homotypic cell-in-cell structures can be regularly found when tumor cells are released into fluid exudates such as ascites or during pleural carcinosis (*Overholtzer and Brugge, 2008*). Although the consequence of entotic invasion is not well understood, the process may contribute to tumor progression by inducing aneuploidy in human cancers (*Krajcovic et al., 2011*). The ultimate outcome of an entotic event also depends on the fate of the invaded cell, which can remain viable or even divide inside or escape from the host cell or undergo vacuolar degradation (*Florey et al., 2010*; *Krajcovic et al., 2011*).

It was previously shown that for a cell to invade into a neighboring cell Rho-dependent signaling and actin are required (*Overholtzer et al., 2007*). However, potential extracellular ligands or cell surface receptors involved in this migratory process are entirely unknown. Furthermore, what type of actin structures and which actin polymerization factor triggers active cell-in-cell invasion in a signal-regulated fashion remained unclear. In this study, we investigated actin-mediated entotic invasion and delineate a signaling pathway downstream of the LPAR2 that ultimately targets the formin mDia1 for polarized actin assembly at the rear of the invading cell to drive cell-in-cell invasion.

## Results and discussion

### Plasma membrane blebbing followed by actin assembly at the rear mediates entotic invasion

To monitor actin assembly during life cell-in-cell invasion over time, we generated MCF10A cells expressing either mCherry- or GFP-LifeAct. Red and green LifeAct-cell populations were mixed and

**eLife digest** Entosis is the invasion of one cell by another and can be observed in aggressive cancers. Although the invading cell is usually killed, the surviving cell is sometimes left with the wrong number of chromosomes. This suggests that entosis may help cancer to progress because cells with an abnormal number of chromosomes are common in cancers.

For entosis to occur, the invading cell must be released from the tissue that surrounds it, so it can move towards and attach to the cell it is about to invade. Very little is currently known about the cellular and molecular events that enable these processes to occur.

Purvanov et al. studied entosis in cells grown in the laboratory and observed that invading cells produce bulges and projections at their rear end for invasion. These projections contain a protein called mDia1. This protein is involved in controlling the growth of the cytoskeleton—the structure that helps cells to both maintain their shape and to move.

Adding the signaling molecule lysophosphatidic acid, which is present in human serum, increased the likelihood that cells would invade others. From this, Purvanov et al. established the identities of the proteins involved in transmitting the lysophosphatidic acid signal that controls mDia1 activity during entosis. Changes to this signaling pathway have been associated with cancer and how it spreads between different organs and its involvement in entosis lends further support to the notion that there may be a link between cell-in-cell invasion and the advancement of cancer.

plated on top of polyHEMA-coated coverslips to prevent matrix adhesion and to induce entotic incidences. Under these conditions, cell-in-cell invasion was confirmed to require ROCK as assessed using the ROCK-inhibitor Y-27632 (*Video 1*) (*Overholtzer et al., 2007*). Interestingly, imaging LifeAct-expressing cells over time, we consistently observed that specifically the actively invading cell displayed extensive blebbing early on during invasion followed by the formation of an actin-rich uropod-like structure at the rear of the invading cell (*Figure 1A*; *Video 2*). Plasma membrane blebbing was highly dynamic under these cell culture conditions with a bleb cycle lasting about 2 min (*Figure 1B*; *Video 3*) and the total number of blebs ranged from 60 to 100 blebs per cell depending on the MCF10A cell size. Notably, blebbing is a frequently observed phenomenon during amoeboid or rounded cancer cell invasion through 3-dimensional collagen requiring ROCK-dependent contractility (*Sahai and Marshall, 2003*; *Kitzing et al., 2007*). The presence of the polarized actin-rich cup at the rear of the entosing cell could be confirmed using phalloidin staining to visualize endogenous actin filaments (*Figure 1C*) or by confocal microscopy of LifeAct-expressing MCF10A cells (*Figure 1—figure supplement 1*).

## LPA promotes cell-in-cell invasion

We noticed that high fetal calf serum (FCS) concentrations enhance entosis (not shown). A ligand known to be present at micromolar concentrations in FCS is lysophosphatidic acid (LPA). Thus, we speculated that LPA might trigger entosis. Indeed, under serum-free conditions addition of LPA efficiently stimulated entotic events of MCF10A cells in a concentration-dependent manner already at nanomolar concentrations (*Figure 1D*). This ligand mediated cell-in-cell invasion was dependent on cell surface receptor activity since the LPA-receptor 1, 2 and 3 inhibitor Ki16425 completely blocked LPA-stimulated entosis (*Figure 1E*). Comparable results were obtained when entosis was triggered by serum (*Figure 1F*). These data identify the soluble extracellular ligand and serum component LPA as a mediator of cell-in-cell invasion.

## The G-protein-coupled receptor LPAR2 mediates entosis

LPA transduces its multiple cellular effects via binding to specific LPA-receptors, which belong to the large superfamily of G-protein-coupled receptors (GPCRs). As there are several different LPA-receptors present in human tissues (*Choi et al., 2010*), we set out to identify the receptor responsible for entotic invasion using an siRNA approach. Interestingly, silencing of LPAR2 resulted in a robust and significant reduction of entotic events, while LPAR5 suppression moderately affected entosis (*Figure 2A*).

To investigate whether LPAR2 is specifically required for the actively invading cell and not for the host cell or both, we applied a two-color entosis assay by stably expressing either GFP- or mCherry-H2B and treated each cell population with siRNA against LPAR2. One phenotypic hallmark characterizing

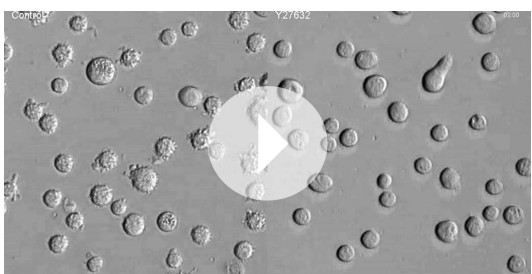

**Video 1**. ROCK activity is required for entosis. Comparison between control and Y27632 (5 μM)-treated MCF10A cells cultured on polyHEMA demonstrating the requirement of ROCK for cell blebbing and cell-in-cell invasion. Time (min) is indicated in the upper right corner.

the host cell from the invading cell during cell-in-cell invasion is the typically half-moon-shaped nucleus (**Figure 1C**; *Overholtzer and Brugge, 2008*). Examination of entotic events using confocal fluorescence microscopy revealed that only cells silenced for LPAR2 failed to actively invade into another, while LPAR2 suppression did not inhibit the host cell during this process (**Figure 2B**). Notably, transient expression of LPAR2 in HEK293 cells significantly triggered entotic invasion (**Figure 2C**), suggesting that disease-associated overexpression or upregulation of LPAR2 as observed in various human cancers (*Goetzl et al., 1999*; *Kitayama et al., 2004*; *Yun et al., 2005*; *Wang et al., 2007*) may be instrumental for entosis.

Next, we assessed the endogenous localization of LPAR2 in entotic cells using immunofluorescence microscopy. Staining of cells with anti-LPAR2 antibodies showed a cortical signal that was distinctively increased at the rear of the invading cell in particular during more progressed phase of entotic invasion (**Figure 2D**), which could be confirmed on transiently expressed Flag-LPAR2 (**Figure 2E**), suggesting that LPAR2-signaling occurs in a defined and more polarized manner. Flag-LPAR2 polarization to the trailing cell rear was independent of downstream actin organization as assessed by addition of latrunculin B, which completely perturbed the cortical actin cytoskeleton (**Figure 2E**, lower panel).

These results establish the LPAR2 as a signal transducer at the cell surface for cell-in-cell invasion.

## Gα$_{12/13}$ and polarized PDZ-RhoGEF activity mediate entotic invasion

LPAR2 can initiate intracellular signaling via coupling to multiple Gα subunits from the G$_i$, G$_q$, and G$_{12/13}$ family of heterotrimeric G-proteins (*Choi et al., 2010*). Silencing various Gα subunits by siRNA revealed that only suppression of Gα$_{12/13}$ effectively and significantly blocked entosis (**Figure 3A**). Consistently, LPAR2-triggered entotic invasion specifically required Gα$_{12/13}$, but not Gα$_{11}$ or Gα$_q$ (**Figure 3B**), clearly demonstrating that LPAR2 signals through G$_{12/13}$ heterotrimeric G-proteins to promote homotypic cell-in-cell invasion. Furthermore, expression of Gα$_{12}$ or of a constitutively active mutant Gα$_{12}$Q/L robustly induced entotic events in the absence of LPA, and this effect was further increased upon addition of 2 μM LPA (**Figure 3C**). Thus, a canonical LPAR2/ Gα$_{12/13}$ module critically mediates entosis.

Gα$_{12/13}$ proteins have been shown to directly relay receptor signal informations by binding to the RGS-domain containing RhoGEFs p115-rhoGEF, LARG, or PDZ-RhoGEF (*Fukuhara et al., 2001*) for activation of the small GTPase RhoA (*Fukuhara et al., 2001*). Therefore, we used siRNA to suppress each of the three RhoGEFs in cells. Interestingly, siRNA-mediated knock-down of PDZ-RhoGEF specifically inhibited entotic events (**Figure 3D**). Furthermore, analyzing ectopically expressed GFP-PDZ-RhoGEF during cell-in-cell invasion revealed a strikingly polarized distribution to the invading cell rear where it strongly colocalized with F-actin as determined by LifeAct (**Figure 3E**). Similar observations were made by staining for phosphorylated myosin-light-chain II (pMLC2) (**Figure 3—figure supplement 1**), a downstream target of the Rho-ROCK pathway, in agreement with previous findings (*Wan et al., 2012*). These data argue that PDZ-RhoGEF promotes localized actin assembly at the cell rear during entotic invasion.

## mDia1 is necessary for entosis downstream of LPAR2

Our LifeAct or non-transfected cell analysis showed that during entosis the invading but not the receiving cells display rigorous membrane blebbing (**Figure 1A**; **Video 2**) reminiscent of bleb-associated cancer cell invasion (*Fackler and Grosse, 2008*). We have shown previously that bleb-associated cancer cell invasion through collagen matrices requires the activity of the Diaphanous formin mDia1 downstream of RhoA (*Kitzing et al., 2007*). We therefore hypothesized that the actin nucleation factor mDia1 may also be involved in entotic invasion. Interestingly, we found endogenous mDia1 to be enriched at the cell rear of the invading cell (**Figure 4A**), suggesting that mDia1 function spatially controls entosis. Indeed, ectopically expressed mDia1-GFP was localized to the actin-rich

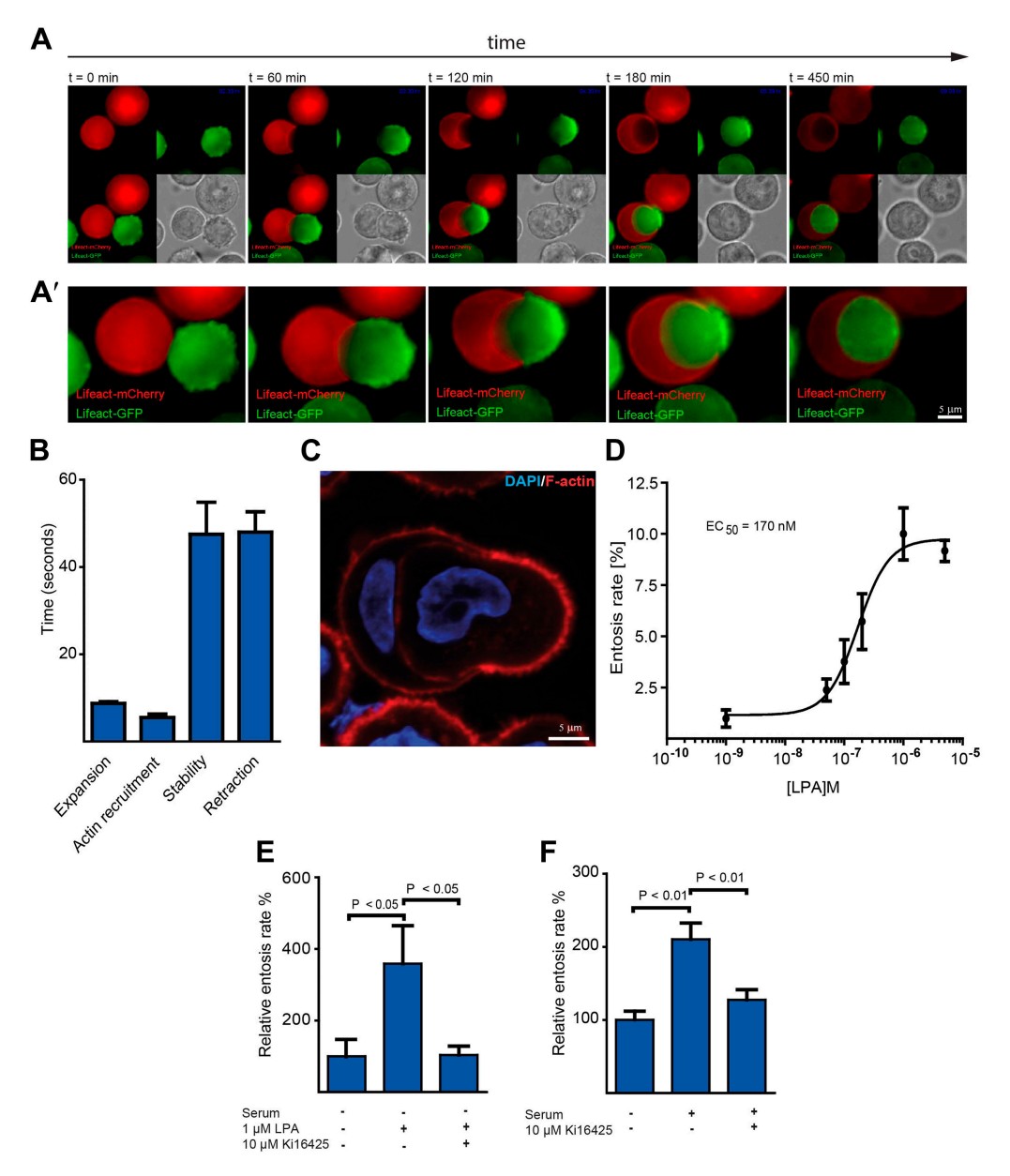

**Figure 1**. Actin dynamics during entotic invasion and stimulation of entosis by LPA. (**A** and **A'**) MCF10A cells expressing LifeAct-mCherry (red) or LifeAct-GFP (green) were monitored over time (***Video 2***) as indicated to visualize actin polymerization during cell-in-cell invasion. Note the specific blebbing activity of the invading cell and the actin-rich structure at the cell rear (green). Differential interference contrast (DIC) is added for each frame. (**B**) Bleb-dynamics were analyzed from eight different live cells expressing LifeAct-GFP, ± SD. (**C**) Entotic MCF10A cells labeled for F-actin using phalloidin (red) and nuclei using DAPI (blue). Scale bar 5 µm. (**D**) Increasing concentrations of LPA stimulate entosis in MCF10A cells under serum-free conditions. (n = 4 ± SD). (**E**) Effects of adding the LPAR1, 2 and 3 receptor blocker Ki16425 on LPA-induced entosis in MCF10A cells (n = 3 ± SEM analyzed by one way ANOVA followed by Dunnett's post-tests compared with LPA-induced group). (**F**) Effects of adding the LPAR1, 2 and 3 receptor blocker Ki16425 on entosis in MCF10A cells after addition of 5% horse serum (n = 4 ± SEM analyzed by one way ANOVA followed by Dunnett's post-tests compared with serum-induced group).

The following figure supplements are available for figure 1:

**Figure supplement 1**. Formation of an actin-rich uorpod-like structure during entotic invasion.

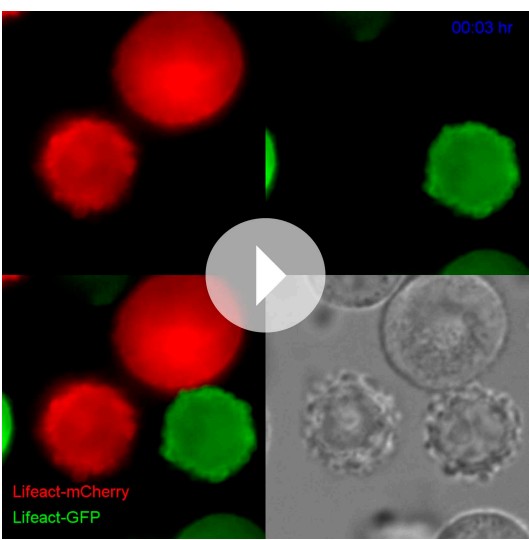

**Video 2**. Actin dynamics during entotic invasion. MCF10A cells expressing LifeAct-mCherry (red) or LifeAct-GFP (green) were monitored over time as indicated (upper right corner) to visualize actin dynamics during cell-in-cell invasion. Video corresponds to **Figure 1A**.

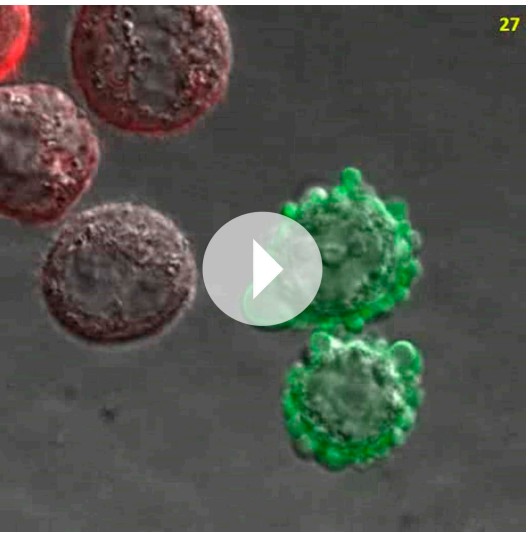

**Video 3**. mDia1 is required for entotic blebbing. MCF10A cells cultured on polyHEMA expressing LifeAct-mCherry (red) or LifeAct-GFP (green) silenced for mDia1 or control respectively were monitored over time as indicated to visualize actin dynamics during blebbing. Video corresponds to **Figure 4—figure supplement 1**.

cup formed in HEK293 cells upon LPAR2 transfection to trigger entotic invasion (**Figure 4B**). This mDia1 localization is also in good agreement with our data showing that the RhoA activator PDZ-RhoGEF is strongly accumulated at the actin-rich cell rear (**Figure 3E**).

Analyzing the cells silenced for mDia1, we found that mDia1 was required for Ezrin-positive bleb formation as well as blebbing (**Figure 4C,D,E**, **Figure 4—figure supplement 1**; **Video 3**), while mDia1 was localized to cellular blebs in control silenced cells (**Figure 4D**). Similarly, blebbing was highly sensitive to LPAR inhibition with Ki16425 (**Figure 4F**; **Video 4 and 5**). Importantly, mDia1 knockdown MCF10A cells were strongly impaired to undergo entotic invasion as compared to control siRNA-treated cells (**Figure 4G,H**; **Video 6**). Under these conditions, when latrunculin B was added just before completion of entosis, we observed the rapid dispersion of the trailing actin-rich cup leading to failure of cell-in-cell invasion (**Figure 4G**; **Video 6**), pointing towards a crucial role for polarized actin assembly during final stages of entosis. Suppression of mDia1 by siRNA treatment also strongly and specifically inhibited LPAR2-triggered entosis (**Figure 4I**) in HEK293 cells, showing that mDia1 is an essential factor that acts downstream of LPAR2 during cell-in-cell invasion.

Homotypic cell-in-cell structures have been reported in metastatic carcinoma cells harvested from exudates or urine samples (**Overholtzer and Brugge, 2008**), corresponding to the non-adhesive experimental culture conditions on hydrogel. It is tempting to speculate that such active invasive and complex process may result in some survival advantage or even represent an escape mechanism for carcinoma cells, although often the inner cell undergoes a cell death process involving components of the autophagy machinery (**Florey et al., 2011**). In this study, we report on a cell surface receptor pathway that facilitates active invasion to produce a cell-in-cell structure. Interestingly, some of these components such as RhoA and mDia1 have been shown to function during rounded cancer cell invasion, which is similarly accompanied by cell blebbing (**Sanz-Moreno and Marshall, 2010**), although this processes can still depend on integrin-based matrix adhesions. Our findings suggest that entotic invasion, although independent of integrins, resembles at least in some aspects that of amoeboid and bleb-dependent motility. Indeed, actin filaments also coincide at the Ezrin-rich uropod in amoeboid blebbing thereby pushing cells through collagen-I (**Lorentzen et al., 2011**) and ezrin is further an essential component for non-apoptotic blebbing (**Charras et al., 2006**). It seems reasonably that Ezrin is potentially also required for entosis as we observed strong Ezrin localization at the invading cell uropod (**Figure 4—figure supplement 1B**), however, its precise role and regulation in this process remains a

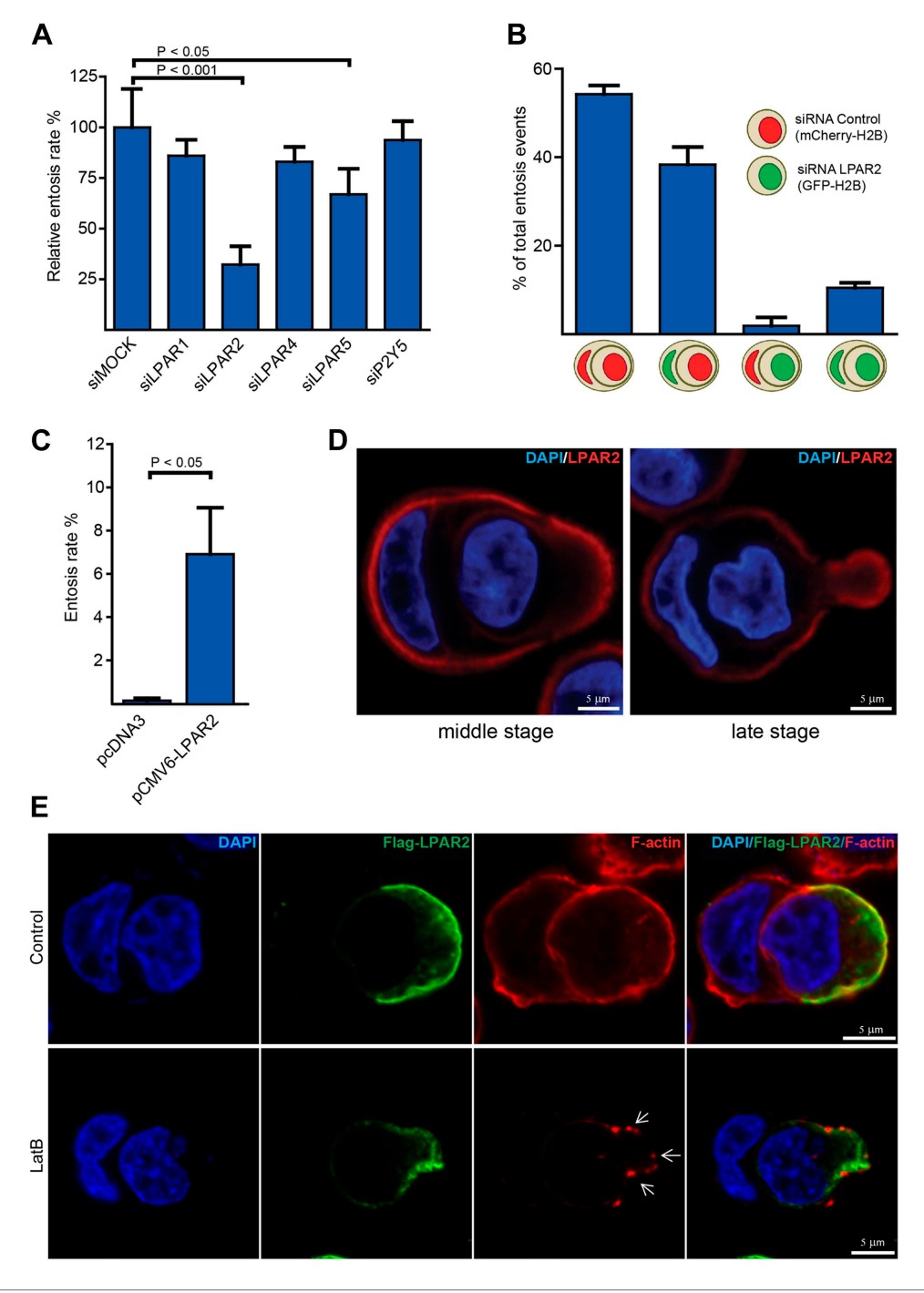

**Figure 2**. LPAR2 triggers invasive motility during entosis. (**A**) MCF10A cells treated with indicated siRNAs for 48 hr were analyzed for entosis (n = 3 ± SD analyzed by one way ANOVA followed by Dunnett's post-tests compared with siMOCK group). (**B**) MCF10A cells stably expressing mCherry-H2B or GFP-H2B were treated with indicated siRNAs before equal cell numbers were mixed and plated to analyze entotic invasion. (**C**) HEK293 cells were transfected with LPAR2 cDNA before analyzation for entosis (n = 3 ± SD, p<0.05, *t* test). (**D**) Immunolabeling of endogenous LPAR2 (red) and nuclei (DAPI) of MCF10A cells fixed at different stages during entosis as indicated. Scale bar 5 μm. (**E**) Immunolabeling of transfected Flag-tagged LPAR2 (green), F-actin (phalloidin, red), and nuclei (DAPI) of invading HEK293 cells undergoing entosis with or without 5 min addition of 100 nM latrunculin B (LatB) before fixation. Arrows indicate disassembled F-actin. Scale bar 5 μm.

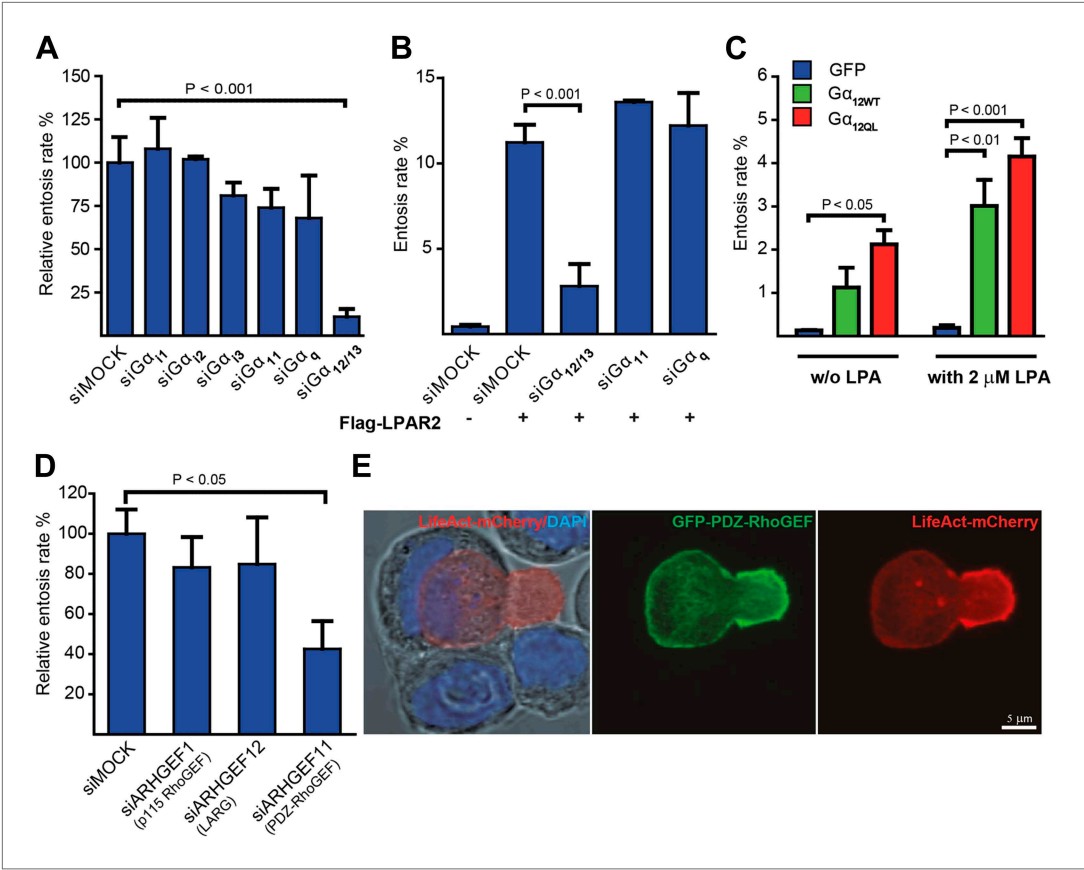

**Figure 3**. Gα$_{12/13}$ and PDZ-RhoGEF are required for entosis. (**A**) MCF10A cells treated with indicated siRNAs for 48 hr were analyzed for relative entosis rates (n = 5 ± SD analyzed by one way ANOVA followed by Dunnett's post-tests compared with siMOCK group). (**B**) HEK293 cells expressing Flag-LPAR2 were treated with indicated siRNAs for 48 hr before analyzing entosis rate (n = 3 ± SD analyzed by one way ANOVA followed by Dunnett's post-tests compared with Flag-LPAR2-expressing siMOCK group). (**C**) HEK293 cells expressing indicated proteins were analyzed for entosis in lipid-depleted medium with or without (w/o) the addition of LPA as indicated. (n = 3 ± SD analyzed by two way ANOVA followed by Bonferroni post-tests). (**D**) MCF10A cells treated with indicated siRNAs for 48 hr were analyzed for entosis (n = 3 ± SD analyzed by one way ANOVA followed by Dunnett's post-tests compared with siMOCK group). (**E**) Localization of GFP-PDZ-RhoGEF (green), DAPI (blue), and LifeAct-mCherry (red) expressed in MCF-7 cells was analyzed by confocal microscopy. Bright-field image merged with DAPI and LifeAct is shown to reveal the cell-in-cell structure (left panel). Note the accumulation of PDZ-RhoGEF at the actin-rich uropod-like structure of the invading cell. Scale bar 5 μm.

The following figure supplements are available for figure 3:

**Figure supplement 1**. Myosin II activity is present at the actin-rich cup at the invading cell rear.

**Figure supplement 2**. Analysis of siRNA treatments.

future task for investigations. The relevance of entosis for tumor progression in vivo is currently unclear. Nevertheless, our data uncover LPA and LPAR2 as important drivers of entotic invasion, both of which are also important factors during cancer metastasis, suggesting that entosis may be a phenomenon associated with advanced malignancy.

## Materials and methods

### Materials

All cell lines were obtained from American Type Culture Collection. MCF10A cells were cultured as described (Debnath et al., 2003). MCF7 and HEK293 cells were cultured in Dulbecco's modified

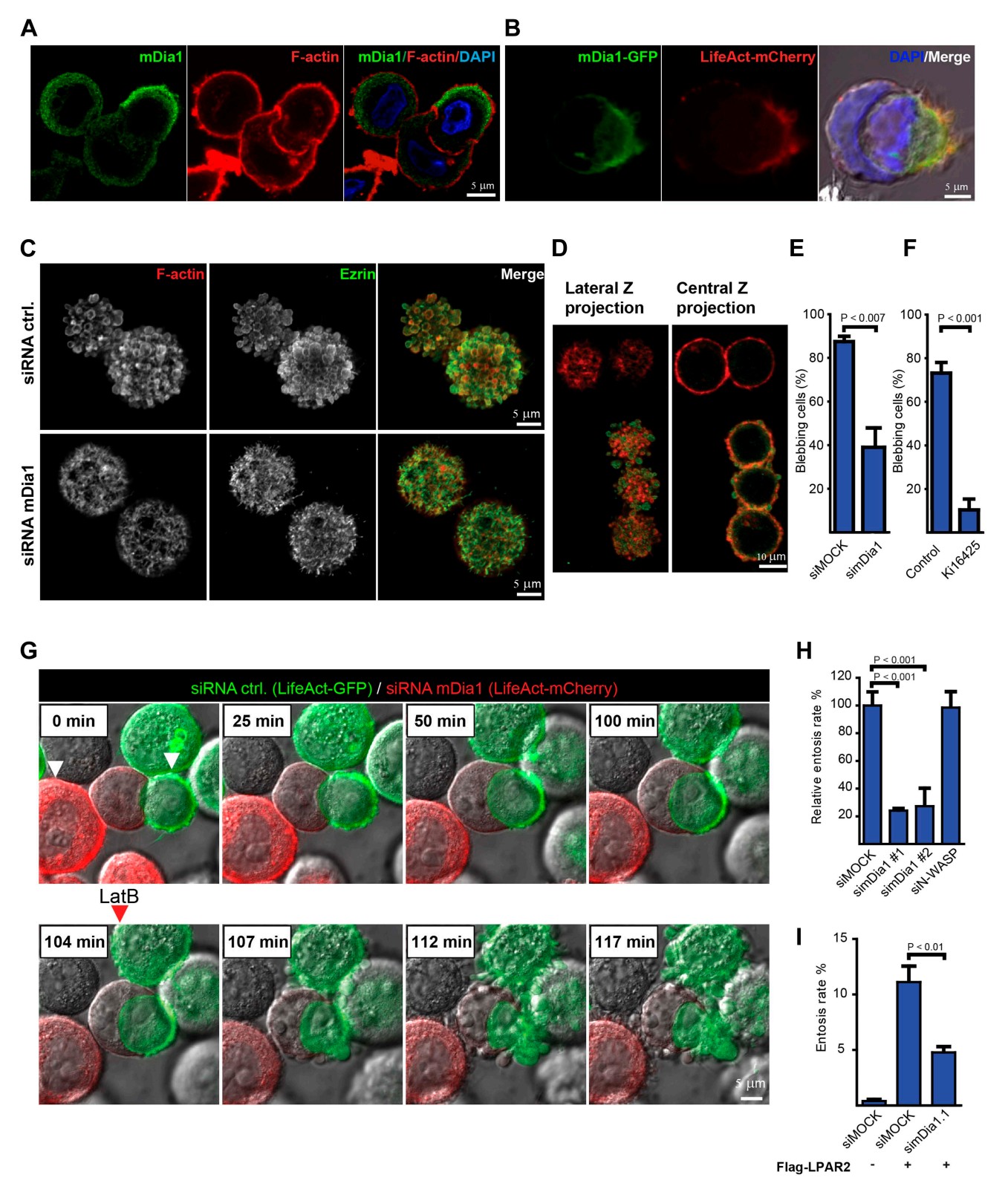

**Figure 4**. The formin mDia1 mediates cell-in-cell invasion downstream of LPAR2. (**A**) Immunolabeling of endogenous mDia1 (green) and phalloidin staining of F-actin (red) of a MCF10A cell undergoing entosis. Nuclei are labeled by DAPI (blue). Scale bar 5 µm. (**B**) Visualization of mDia1-GFP (green)

*Figure 4. Continued on next page*

*Figure 4. Continued*

and mCherry-LifeAct (red) localization at the invading cell rear in fixed and non-permeabilized HEK293 cells co-transfected with LPAR2 to trigger cell-in-cell invasion events. Merged image including bright-field and DAPI (blue) is shown in the right panel. Scale bar 5 µm. (**C**) Immunolabeling of endogenous Ezrin (green) and F-actin (red) of control and mDia1 siRNA-treated MCF10A cells. (**D**) MCF10A cell population after incomplete siRNA treatment against mDia1 showing mDia1 knockdown of the upper two cells (red only) and endogenous mDia1 detection of the lower three cells were labeled for mDia1 (green) and F-actin (red). Note the presence of mDia1 on cellular blebs, while the two upper mDia1-negative cells fail to bleb. 2 frames are shown from a confocal z-scan using a LSM 700 (Zeiss). (**E**) MCF10A cells treated with indicated siRNAs were analyzed for the number of blebbing cells (n = 3 ± SD, p<0.007, *t* test). (**F**) MCF10A cells pretreated for 40 min with 20 µM of the LPAR inhibitor Ki16425 before analysis of the number of blebbing cells (n = 3 ± SD, p<0.001, *t* test). (**G**) MCF10A cells expressing LifeAct-GFP (green) or LifeAct-mCherry (red) silenced for control or mDia1 respectively. White arrowheads in the first frame indicate red (siDia1) and green (siMOCK) cell in contact with a host cell. Red arrowhead indicates addition of 100 nM Latrunculin B (LatB) at time frame 104 min. (**H**) MCF10A cells treated with indicated siRNAs for 48 hr were analyzed for entosis (n = 3 ± SD analyzed by one way ANOVA followed by Dunnett's post-tests compared with siMOCK group). (**I**) HEK293 cells expressing Flag-LPAR2 to trigger cell-in-cell invasion events were treated with indicated siRNAs for 48 hr before analyzing entosis rates (n = 3 ± SD analyzed by One way ANOVA followed by Dunnett's post-tests compared with Flag-LPAR2 expressing siMOCK group).

The following figure supplements are available for figure 4:

**Figure supplement 1**. mDia1 is required for blebbing.

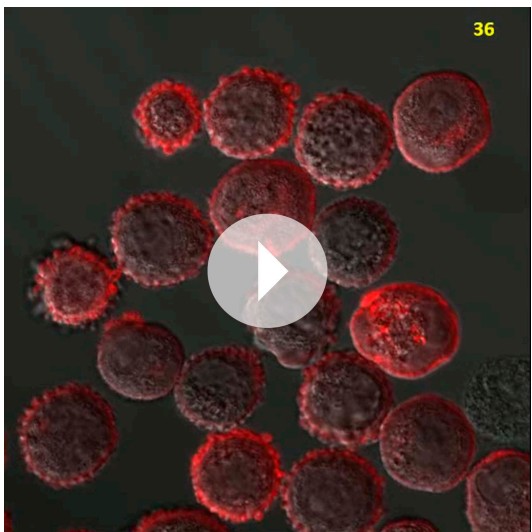 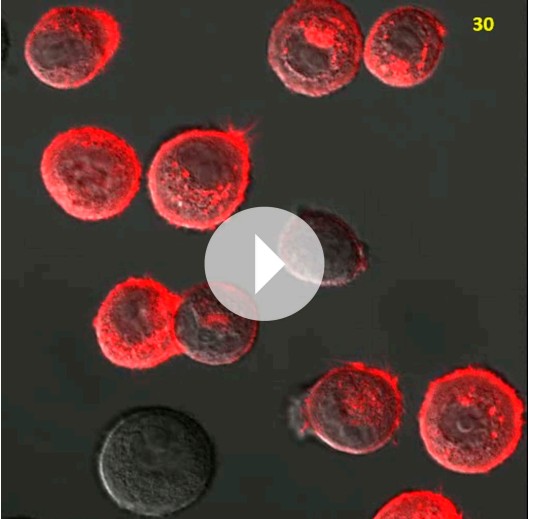

**Video 4**. Actin dynamics during blebbing. MCF10A cells expressing LifeAct-mCherry (red) were monitored over time as indicated (upper right corner) to visualize blebbing cells. Video corresponds to quantifications in *Figure 4F* as control to *Video 5*.

**Video 5**. Effects of LPAR inhibition on cell blebbing. MCF10A cells expressing LifeAct-mCherry (red) were treated with the LPAR inhibitor Ki16425 and monitored over time to visualize effects on cell blebbing. Video corresponds to quantifications in *Figure 4F*.

Eagle's medium (DMEM) plus 10% heat inactivated FBS. Cell Dissociation Buffer, enzyme-free, PBS-based (Life Technologies). Oligonucleotides of small interfering RNA (siRNA) for LPA receptors, G protein alpha subunits, and RhoGEFs were synthesized by QIAGEN. Oligonucleotides of siRNA for mDia1 were purchased from IBA GmbH. LPA (1-Oleoyl Lysophosphatidic Acid) and EDG family inhibitor Ki16425 were purchased from Cayman Chemical Company. Poly (2-hydroxyethyl methacrylate) (PolyHEMA) was purchased from Polysciences Inc. Antibodies against EDG4 were from Assay Biotechnology; LPAR receptors, Ezrin and LARG from Santa Cruz Biotechnology; PDZ-RhoGEF from IMGENEX; pMLC2 from Sigma and mDia1 from BD Biosciences. pCMV6-XL5 LPAR2 expression vector were purchased from OriGene (SC117226). pWPXL-based lentiviral expression vectors for H2B, LPAR2, Gα12, and Gα12Q/L were generated using standard PCR-based procedures or in the case of LifeAct-GFP were a kind gift from Oliver Fackler. Delipidized FCS was from Bio&SELL e.K.

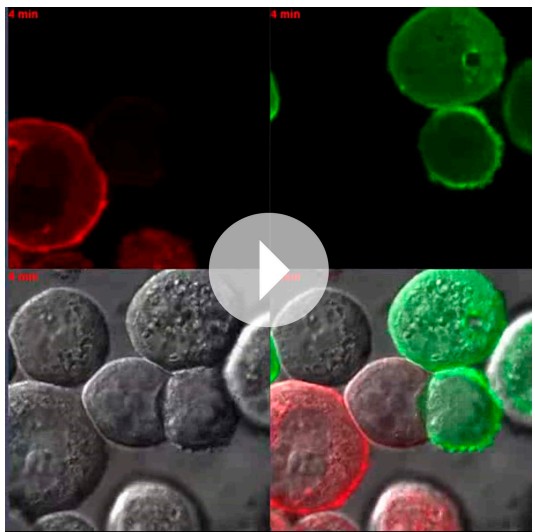

**Video 6**. mDia1 is required for cell-in-cell invasion. MCF10A cells expressing LifeAct-GFP (green) or LifeAct-mCherry (red) silenced for control or mDia1 respectively were monitored over time (indicated in each pabel) for entosis. Latrunculin B (100 nM) was added before full completion of cell-in-cell invasion at time point 104 (min) to dissolve the trailing actin-rich uropod-like structure. Video corresponds to *Figure 4G*.

## siRNA knockdown experiments

MCF10A or MCF7 cells were transiently transfected with 10–50 nM siRNA oligonucleotides by using INTERFERin (Polyplus Transfection Inc) and knockdown was quantified by qPCR or confirmed by Western analysis (*Figure 3—figure supplement 2*). The following FlexiTube siRNA (QIAGEN) were used: GNAI1 SI00032256; GNAI2 SI02780505; GNAI3 SI00088942; GNAQ SI02780512; GNA11 SI0265947; GNA12 SI00096558; GNA13 SI00089761; GNA14 SI00062321; ARHGEF1 SI00302680; ARHGEF11 SI00108129; ARHGEF12 SI04352278; AKAP13 SI02224173; EDG1 SI00376229; EDG4 SI00067494; EDG7 SI00097545; GRP23 SI00075292; GPR92 SI00126231; P2RY5 SI00081116; The following siRNAs from IBA GmbH were used: mDia1.1 #97082N/97083N; mDia1.2 #97273/274N.

## Entosis assays

For general quantitative assessments monolayer cells were trypsinized (or subjected to Cell Dissociation Buffer in case of HEK293) to obtain a single-cell suspension before plating on Ultra Low Cluster Plate (costar cat. 3473) at densities of 300.000–400.000 cells per well. Cells processed for immunostaining were fixed in suspension by adding 1:1 vol/vol 8% formalin for 10 min. Cells were then rehydrated in PBS and centrifuged on to 12-mm cover slips using a Cytospin Cytofuge12 at 1500 rpm for 4 min by high acceleration. Fixed samples were washed in PBS and PBST and blocked in Blocking Buffer (PBS, 0.2% Triton X-100, 5% Goat Serum, 0.1% BSA, 0.4% Glycerol) before antibody addition. Nuclei were stained with DAPI and F-actin using Alexa-647-phalloidin or Alexa-488-phalloidin.

## Microscopy

Time-lapse microscopy was performed on dishes coated with PolyHEMA as described (*Overholtzer et al., 2007*). Images were obtained using a Nikon Eclipse-Ti equipped with Perfect Focus under humidified conditions at 37°C (Tokai hit stage top incubator) using a Nikon 40x oil objective. Confocal microscopy was performed using a 63x objective on a ZEISS LSM-700. For all entosis quantifications more than 600 cells were counted per coverslip.

## Acknowledgements

We thank Oliver Fackler for lentiviral LifeAct-mCherry and Andrea Wüstenhagen for technical assistance. MH was supported by a Mildred-Scheel-Doktorandenprogramm fellowship (# 110405) from the Deutsche Krebshilfe e.V. We thank laboratory members for discussions and the SFB 593 and SFB-TR17 for funding.

## Additional information

### Funding

| Funder | Grant reference number | Author |
|---|---|---|
| Deutsche Forschungsgemeinschaft (DFG) | SFB 593, SFB-TR17 | Robert Grosse |
| Deutsche Krebshilfe e.V. | 110405 | Manuel Holst |

The funders had no role in study design, data collection and interpretation, or the decision to submit the work for publication.

## Author contributions
VP, JK, Conception and design, Acquisition of data, Analysis and interpretation of data; MH, CB, Conception and design, Acquisition of data, Analysis and interpretation of data, Drafting or revising the article; RG, Conception and design, Analysis and interpretation of data, Drafting or revising the article

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
