## [Decision Letter]

Thank you for sending your work entitled “LPAR2 couples G_12/13_-mediated signaling and polarized actin assembly for homotypic cell-in-cell invasion” for consideration at *eLife.* Your article has been favorably evaluated by a Senior editor and 3 reviewers, one of whom is a member of our Board of Reviewing Editors.

The Reviewing editor and the other reviewers discussed their comments before we reached this decision, and the Reviewing editor has assembled the following comments to help you prepare a revised submission.

The reviewers agreed that entosis (cell-in-cell invasion) is an interesting, and potentially important process in cancer, in which cells engulf each other. While the phenomenon itself is well-described, and a role for actin and autophagy has been shown to be important for events after engulfment, little is known about the receptor(s) that recognize cells for entosis. They noted that you describe the identification for a receptor and downstream pathway for entosis. You identify a requirement for LPA, and this led them to identify, by siRNA and ectopic expression, the G-protein coupled receptor LPAR2 as the critical receptor. You go on to show that the downstream pathway involves G12/13 family GPCRs, PDZ-RhoGEF and mDia1 that induce actin remodeling in the rear uropod of the engulfing cell.

The 3 reviewers found the work interesting and potentially important. However, all raised important questions about the relationship(s) between blebbing, RhoA signaling, mDia1 function, trailing uropod formation/function and entosis. These concerns will need further work and, if completed, another round of consultation. The major concerns are summarized below:

1) Figure 1 shows the effect of the LPAR inhibitor Ki16425 in the presence of added LPA, but this should also be tested in the presence of serum to show that the LPAR/LPA link is the “only” critical component in whole serum as well.

2) An increase in blebbing in the invading cells is mentioned throughout in the manuscript, and is used as a rationale for examining mDia1. The blebbing phenotype should be better characterized (number of blebs/membrane protrusions over time, changes in ERM protein expression/localization, etc.), and it would be important to test whether inhibiting signaling and knockdown of actin regulatory protein (particularly PDZ-RhoGEF and mDia1) affect entosis-associated blebbing.

3) What is the role of the uropod in entosis? Is it required for entosis, or is it a by-product of the entosis process? Linking the role of the uropod and the Rho/mDia1 signaling pathway could be addressed by testing whether inhibition of mDia1 (or RhoA signaling) blocks uropod formation and either the initiation of entosis (i.e., LPAR localization and binding to the recipient cell) or the final stage(s) of engulfment.

4) Is the localization of LPAR to the uropod dependent on an intact actin cytoskeleton, and is its distribution affected by the knockdown of expression of G12/13, RhoGEF or mDia1, or inhibition of ROCK? This information is important in the context of showing whether LPAR localization is independent or dependent of downstream effectors and actin organization.

5) Roles for Gα12/13 and PDZ-RhoGEF in entosis (Figure 3) are intriguing, but more work is needed to rigorously establish whether this leads to Rho activation. Activation could be measured using a Rho FRET sensor, and tested if and where RhoA is active at sites of entosis under both activating (LPA addition, LPAR overexpression, Gα12QL) and inhibiting (siRNA knockdown of Gα12/13, PDZ-RhoGEF) conditions.

6) The cellular localization of mDia1, both endogenous and GFP-tagged, appears cytoplasmic in the invading cell (Figure 4), and it is difficult to see the enrichment in the trailing uropod as the authors claim: for example, mDia1 appears to be excluded from the F-actin rich band in the trailing uropod (Figure 4). The effects of mDia1 knockdown on actin cytoskeleton and membrane blebbing should be quantitatively analyzed to test whether mDia1 is inducing these changes.

Other concerns:

1) The percent knockdown of different LPAR isoforms (Figure 2), GPCR isoforms (Figure 3), RhoGEFs (Figure 3) and mDia1 (Figure 4) should be shown by western blot so that the conclusion that one isoform is required and not others is based on equal degrees of loss expression.

2) It should be clearly stated in the text that localization of specific proteins (e.g. mDia in Figure 4) is by monitoring ectopically-expressed, GFP-labeled fusion proteins.

3) Statistical analysis:

- Student's t-test should only be employed when 2 conditions are being compared (e.g. Figure 2). Where there are more than 2 groups, appropriate tests (e.g. one-way ANOVA followed by post-hoc test) should be used.

- Wherever possible, the true variance of the control group (e.g. Figure 2) should be included, even when normalized to 100%. Including the variance would allow for robust statistical analysis, which currently is not the case since the variance has been removed.

- The number of replicates should be listed in the figure legend for Figure 1, and statistics provided for Figure 1.

4) The LPAR localization studies would benefit from the inclusion of general membrane and cytoplasmic markers (e.g. membrane-targeted and cytoplasmic GFP). This would help to demonstrate that LPAR2 is indeed clustered at the trailing edge. Examining the localization another LPAR (4 or 5) would be useful to show specificity of LPAR1 localization.

---

## [Author Response]

*1)*
Figure 1
*shows the effect of the LPAR inhibitor Ki16425 in the presence of added LPA, but this should also be tested in the presence of serum to show that the LPAR/LPA link is the “only” critical component in whole serum as well*.

We agree that this is an important point that we have previously overlooked and have performed the suggested experiments now included in Figure 1, showing a strong and significant requirement for LPAR also under serum-induced entosis.

*2) An increase in blebbing in the invading cells is mentioned throughout in the manuscript, and is used as a rationale for examining mDia1. The blebbing phenotype should be better characterized (number of blebs/membrane protrusions over time, changes in ERM protein expression/localization, etc.), and it would be important to test whether inhibiting signaling and knockdown of actin regulatory protein (particularly PDZ-RhoGEF and mDia1) affect entosis-associated blebbing*.

The reviewers rightly point out that blebbing in the invading cell is observed and described throughout the manuscript but may have been less well characterized in the previous version. We thank the reviewers for these comments in particular as the blebbing activity observed here occurs under integrin-independent cell culture conditions and we have made efforts to address these issues with further experiments. We now provide data in Figure 1 on the bleb dynamics as requested (see also Video 3). The average number of blebs per cell is now mentioned in the text. We have further, as requested, examined Ezrin (ERM) expression/localization and blebbing after knockdown of the actin-regulatory protein mDia1 and found that overall Ezrin expression appears to be unaffected, however, blebbing and hence Ezrin localization is severely inhibited and disturbed in mDia1 knockdown cells. These data are now presented in Figure 4 C, D, E, Figure 3—figure supplement 2, and Videos 3 and 4.

Together these data show that mDia1 is essential for proper and productive blebbing and that the blebbing of the invading cell is in fact required for the invasion process. Thus, if blebbing is perturbed no entotic invasion can occur and hence no uropod can be formed by the invading cell later in the process.

*3) What is the role of the uropod in entosis? Is it required for entosis, or is it a by-product of the entosis process? Linking the role of the uropod and the Rho/mDia1 signaling pathway could be addressed by testing whether inhibition of mDia1 (or RhoA signaling) blocks uropod formation and either the initiation of entosis (i.e., LPAR localization and binding to the recipient cell) or the final stage(s) of engulfment*.

We agree that these are exciting questions. As mentioned above in point 2, we cannot address the requirement of Rho/mDia1 for uropod formation since inhibition of these factors interferes with blebbing and entotic invasion or intitiation early on before uropod formation during later stages of entosis. Thus, these factos are also required for bleb-associated initiation of entosis as also shown in Figure 4 (Video 4), which illustrates that si-mDia1 (red) treated cells do not invade although they still make contacts to a neighboring cell, which becomes entosed by a control cell (green). Thus it seems that the entire actin-driven processes is under the control of the Rho/mDia1 pathway such as blebbing for initiation and completion of entotic invasion. Along these lines, it can be noted that the actin-rich uropod-like structure is also actively blebbing (Figures 1 and 4, Videos 2 and 4). Furthermore, we do find good accumulation of endogenous or mDia1-GFP specifically in the invading cell rear Figure 4.

In addition, we find strong signal accumulation of p-MLC2 at the actin rich cup/uropod, clearly indicating that polarized Rho-actin signalling is occurring (Figure 3—figure supplement 1). This supports a model in which this trailing actin structure provides Rho/actin-dependent force to push the remaining cell body inside the other. In fact, when we add LatB just before completion of entosis we halt invasion due to cortical actin relaxation (Figure 4 and Video 4 from time frame 104 minutes).

Thus, although we cannot instantly inhibit specifically Rho or mDia1 late during entotic uropod formation at present (efforts trying to add the generic and unspecific formin inhibitor smiFH2 have failed at least in our hands) our overall data support the notion that the uropod-like structure is necessary for completion of entotic invasion.

*4) Is the localization of LPAR to the uropod dependent on an intact actin cytoskeleton, and is its distribution affected by the knockdown of expression of G12/13, RhoGEF or mDia1, or inhibition of ROCK? This information is important in the context of showing whether LPAR localization is independent or dependent of downstream effectors and actin organization*.

We have addressed this by studying the localization of Flag-LPAR2 during entosis after addition of Latrunculin B (5 minutes before fixation) to dissolve the F-actin cytoskeleton during late entotic invasion and observed that LPAR polarization to the cell rear is independent of downstream actin organization despite profound perturbation of the cortical actin cytoskeleton (now shown in Figure 2, lower panel).

*5) Roles for Gα12/13 and PDZ-RhoGEF in entosis (*Figure 3*) are intriguing, but more work is needed to rigorously establish whether this leads to Rho activation. Activation could be measured using a Rho FRET sensor, and tested if and where RhoA is active at sites of entosis under both activating (LPA addition, LPAR overexpression, Gα12QL) and inhibiting (siRNA knockdown of Gα12/13, PDZ-RhoGEF) conditions*.

We regret that we are at present unable to perform the requested experiments using a Rho-FRET sensor since the establishment of such more complex assay in an already difficult to study cell system such as entosis, including that we have no experience with FRET in our lab so far, would take us very likely more than 6 month with an unclear success outcome. We can only say that we will try to establish this procedure in the future, as these are certainly relevant aspects. Nevertheless, we performed phospho-myosin light chain 2 (p-MLC2) stainings and find that p-MLC2 appears to be strongly enriched at the actin-rich cup now shown as Figure 3—figure supplement 1.

*6) The cellular localization of mDia1, both endogenous and GFP-tagged, appears cytoplasmic in the invading cell (*Figure 4*), and it is difficult to see the enrichment in the trailing uropod as the authors claim: for example, mDia1 appears to be excluded from the F-actin rich band in the trailing uropod (*Figure 4*). The effects of mDia1 knockdown on actin cytoskeleton and membrane blebbing should be quantitatively analyzed to test whether mDia1 is inducing these changes*.

We now provide what we hope is more convincing evidence for mDia1 enrichment at the trailing uropod (new Figure 4) as well as at the sites of blebbing (Figure 4). We also, as suggested, quantified the effects of mDia1 knockdown on cell blebbing, now shown in Figure 4.

Other concerns:

*1) The percent knockdown of different LPAR isoforms (*Figure 2*), GPCR isoforms (*Figure 3*), RhoGEFs (*Figure 3*) and mDia1 (*Figure 4*) should be shown by western blot so that the conclusion that one isoform is required and not others is based on equal degrees of loss expression*.

We now provide these data as suggested in Figure 4—figure supplement 1 and for mDia1 Figure 3—figure supplement 2.

*2) It should be clearly stated in the text that localization of specific proteins (e.g. mDia in*
Figure 4*) is by monitoring ectopically-expressed, GFP-labeled fusion* proteins.

We apologize if this was not clearly indicated and we have now corrected that accordingly.

3) Statistical analysis:

*- Student's t-test should only be employed when 2 conditions are being compared (e.g.*
Figure 2*). Where there are more than 2 groups, appropriate tests (e.g. one-way ANOVA followed by post-hoc test) should be used*.

*- Wherever possible, the true variance of the control group (e.g.*
Figure 2*) should be included, even when normalized to 100%. Including the variance would allow for robust statistical analysis, which currently is not the case since the variance has been removed*.

*- The number of replicates should be listed in the figure legend for*
Figure 1*, and statistics provided for*
Figure 1.

We apologize for these mistakes and have now corrected them and indicted the details in the according figure legends.

*4) The LPAR localization studies would benefit from the inclusion of general membrane and cytoplasmic markers (e.g. membrane-targeted and cytoplasmic GFP). This would help to demonstrate that LPAR2 is indeed clustered at the trailing edge. Examining the localization another LPAR (4 or 5) would be useful to show specificity of LPAR1 localization*.

We now show in the new Figure 2 that LPAR2 localization to the trailing edge appears to be specific as at earlier stage of entosis when cortical F-actin can still be uniformly distributed LPAR2 is already polarized. This further suggests that LPAR2 polarized localization can precede downstream actin assembly such as for actin-rich trailing cup formation.